# Fish Size Structure as an Indicator of Fish Diversity: A Study of 40 Lakes in Türkiye

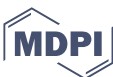

Thomas Boll [1,2], Şeyda Erdoğan [1,3], Ümmühan Aslan Bıçkı [4], Nur Filiz [1,5], Arda Özen [1,6], Eti Ester Levi [1,7], Sandra Brucet [7,8,9], Erik Jeppesen [1,5,7,10] and Meryem Beklioğlu [1,5,*]

1  Limnology Laboratory, Department of Biological Sciences, Middle East Technical University, 06800 Ankara, Türkiye
2  COWI A/S, Jens Chr. Skous Vej 9, 8000 Aarhus, Denmark
3  Department of Biology, Faculty of Science and Art, Yozgat Bozok University, 66900 Yozgat, Türkiye
4  Department of Forest Protection, Wildlife and Protected Areas, Southwest Anatolia Forest Research Institute, 07010 Antalya, Türkiye
5  Centre for Ecosystem Research and Implementation, Middle East Technical University, 06800 Ankara, Türkiye
6  Department of Forest Engineering, Çankırı Karatekin University, 18200 Çankırı, Türkiye
7  Department of Ecoscience and WATEC, Aarhus University, 8000 Aarhus, Denmark
8  Department of Environmental Sciences, University of Vic, 08500 Vic, Spain
9  ICREA, Catalan Institution for Research and Advanced Studies, 08010 Barcelona, Spain
10 Sino-Danish Centre for Education and Research (SDC), Beijing 100380, China
*  Correspondence: meryem@metu.edu.tr; Tel.: +90-312-210-51-55

**Abstract:** Body size is a master trait in aquatic ecosystems to complement traditional taxonomic diversity measures. Based on a dataset of fish communities from 40 Turkish lakes covering a wide environmental gradient and continental to dry cold steppe to Mediterranean climates, we elucidated the key variables controlling size diversity, geometric mean length and number of size classes in the fish community. We further examined how these three size measures were related to species diversity and species richness. A GLM analysis revealed that both size diversity and the number of sizes were strongly related to taxonomic diversity and richness. Furthermore, fish size diversity decreased with decreasing annual precipitation, while the number of size classes increased with increasing lake area but decreased with increasing salinity. The geometric mean length of fish decreased with total nitrogen and increased with altitude. The inter-relatedness between the number of size classes and lake area suggests an increase in fish niches with increasing ecosystem size, while fish are smaller and have fewer size classes in lakes with higher salinity. We conclude that size measures provide valuable integrating information on lake fish diversity; thus, they may complement, but not replace, more traditional taxonomic fish measures.

**Keywords:** size diversity; lake fish community; geometrical length; nutrients; salinity

## 1. Introduction

Body size is a master trait in ecology that affects how physiology [1] correlates with trophic position [2] and determines competitive ability and predator–prey interactions [2–4]. Body size, thus, has several functional attributes and plays a key role in structuring communities, trophic interactions and food webs [2,5]. Size diversity (based on individual body size) of a community might, therefore, be a proxy for functional diversity [6–11].

The size structure of fish communities in lakes has been shown to be influenced by several factors including temperature [12,13], lake size [14], resource availability and eutrophication [15], and fishery [16]. For example, fish tend to be small-bodied in areas with high ambient temperature, i.e., areas at low latitude and low altitude [12,13,17], which is in accordance with the temperature–size rule [18]. Larger lakes also often support larger-sized fish [14] and longer fish food chains [19]. Moreover, larger lakes are expected to support

more species due to their more diverse habitats and potentially more niches [20], and a larger lake size may also allow the co-existence of fish covering a wide size range [21].

The relationship between species richness and size diversity may help to understand the mechanisms shaping community structure, while integrating variation at both intraspecific and interspecific levels [22–24] that may be neglected by measuring taxonomic diversity alone [6,11]. However, only very few studies have evaluated this relationship in freshwater fish assemblages [11]. The slope of the species diversity and size diversity relationship indicates the rate of increasing new body sizes: a steep slope means that the diversity of body sizes increases fast with species diversity (i.e., high congruency between species and size diversity), whereas a shallower slope implies an increasing overlap in size while species accumulate (i.e., weak correspondence between species and size diversity). At the European scale, the size diversity of lake fish communities was found to be similarly high for different levels of species diversity (i.e., a positive relationship between species diversity and size diversity was observed but with a shallow slope), which means that the size diversity is not a strong surrogate for species diversity [11]. Nevertheless, the relationship between species diversity and size diversity changed across the continent, with the greatest mismatch occurring in northern Europe and higher congruence towards lower latitudes where fish species diversity is disproportionately high.

Freshwater ecosystems, despite covering less than 1% of the surface of the planet, support up to 6% of all known species [25] and almost 9.5% of all known animal species [26], making them one of the most diverse ecosystems on the planet. Furthermore, the levels of endemism among freshwater species, especially for fish species, are remarkably high. For instance, of the fish species assessed for the freshwater ecoregions of the world, over half are confined to a single ecoregion [27]. However, biodiversity loss in freshwaters is much higher compared to the marine and terrestrial realms. The International Union for Conservation of Nature (IUCN) Red List of Threatened Species classifies nearly 30% of all freshwater fishes as threatened [28]. Freshwater fish species appear to be more susceptible. This rapid decline of freshwater species, especially fish, requires a rapid, reliable, yet easy assessment with in-depth information about the community structure, especially in warm latitudes where the diversity of fish is very high.

Here, we investigated how different size measures (size diversity, geometric mean length and number of size classes) of lake fish assemblages are related to fish taxonomic diversity measures along a latitudinal gradient of lakes in Türkiye, as well as how they may deviate with environmental conditions. We hypothesized a positive relationship between size diversity and species diversity. However, as some of our study lakes were located in arid to semi-arid regions, we expected that temperature and precipitation would also play important roles in the structuring body size (e.g., fish being smaller in areas with high temperatures and size diversity being lower in areas with extreme temperatures and low precipitation where salinity may become a predominant driver (due to a harsh environment and fewer species)). At the same time, we also expected a positive correlation with the lake area due to its higher potential of more niches and higher species richness.

## 2. Materials and Methods

### 2.1. Study Area

We included the 28 Anatolian lakes used in the European scale study by Brucet et al. [10] and added another 12 Anatolian lakes to the dataset. Our 40 study lakes are all natural, distributed across the mid and western half of Anatolia, Türkiye, spanning over five latitudes exhibiting large differences in spatial, climatic and environmental conditions (Figure 1, Table 1). The lakes are located in mountainous areas, on the Central Anatolian Plateau as well as in the lowland along the coasts of the Black Sea and the Aegean Sea, with altitude ranging from 0.3 to 1423 m.a.s.l. (Table 1). The climatic features of the study lakes vary from hot and dry summers in the coastal lowlands, a warm temperate/Mediterranean climate in the western and southern parts to the cold steppe of the central part of the Anatolian Plateau where summers are warm and dry [29]. The Northern Anatolian Mountains,

situated in the northern part of the study area along and close to the coastline of the Black Sea, have a fully humid climate with warm summers in the highland and hot summers in the coastal lowland [29]. Annual average air temperature and total annual precipitation (Table 1) were calculated for each lake from data provided by www.worldclim.org (accessed on 10 November 2013). Seasonality, measured as the temperature difference between mean air temperature in the coldest (January) and the warmest (July) months, ranged from 15.8 °C to 22.4 °C, the lowest seasonality occurring in lowland lakes.

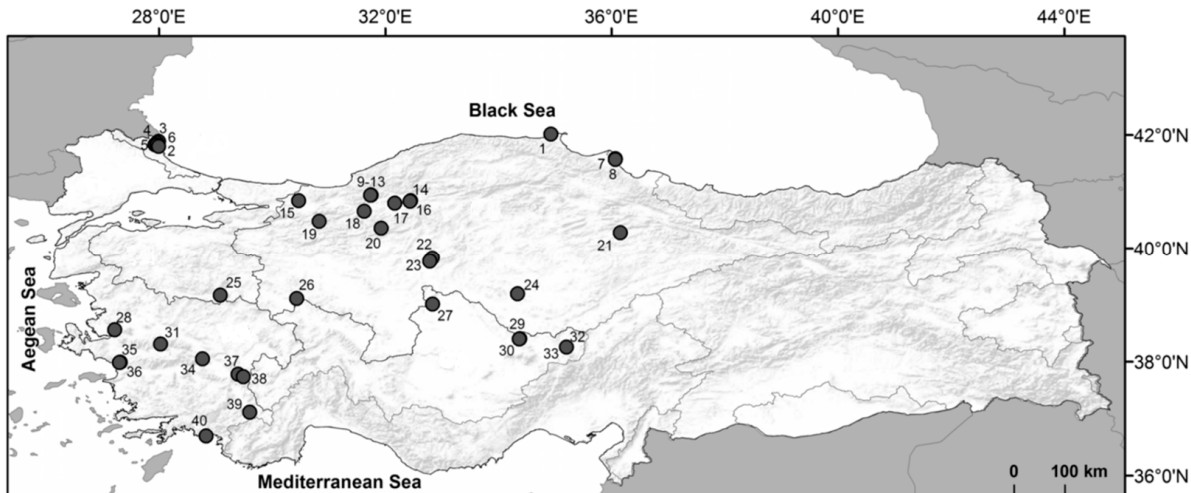

**Figure 1.** Map showing Türkiye and the distribution of the 40 lakes included in the study; the names of the lakes are: (1) Sarıkum, (2) Erikli, (3) Mert, (4) Pedina, (5) Hamam, (6) Saka, (7) Gıcı, (8) Tatlı, (9) Serin, (10) Büyük, (11) Derin, (12) Ince, (13) Nazlı, (14) Koca, (15) Poyrazlar, (16) Keçi, (17) Gerede, (18) Gölcük (B), (19) Çubuk, (20) Karagöl (B), (21) Kaz, (22) Eymir, (23) Mogan, (24) Seyfe, (25) Gölcük (S), (26) Emre, (27) Gök Göl, (28) Karagöl (İ), (29) Kayı, (30) Balıklı, (31) Gölcük (Ö), (32) Eğri, (33) Sarp, (34) Yayla, (35) Barutçu, (36) Gebekirse, (37) Saklı, (38) Karagöl (D), (39) Gölhisar, (40) Baldımaz. Sixteen of the study lakes were not hydrologically connected to any other sampled lake. The remaining 24 lakes were hydrologically connected to one or more sampling lakes. These include the lakes numbered 2 and 3; 7 and 8; 9–13; 14 and 16–17; 22 and 23; 32 and 33; 35 and 36, are connected.

Lake trophic state ranged from mesotrophic to hypereutrophic with large differences in total nitrogen (TN), total phosphorous (TP) and chlorophyll a concentrations as well as in Secchi disc depth (Table 1). Submerged macrophytes, measured as the percentage of the lake's total volume inhabited by macrophytes (PVI), reached up to 80% but with a median PVI across all lakes of only six. Also, salinity varied among lakes—six lakes had salinities > 3.0‰, the highest salinity being 14.5‰ and <1.5‰ in the remaining lakes (Table 1). Further details about the ecology of the lakes can be found in Beklioğlu et al. [30].

**Table 1.** A summary of the spatial, climatic, and environmental conditions characterizing the 40 lakes included in the study. Physical, chemical and biological data were collected in a snapshot sampling conducted in each lake at the time of the fish survey (July and August). The sampling procedure and environmental data are described in Boll et al. [31]. and Levi et al. [32]. Abbreviations were added in the parentheses).

| Variables | Median | Minimum | Maximum |
|---|---|---|---|
| Latitude (Lat) | | 36°70′ | 42°01′ |
| Longitude (Lon) | | 27°22′ | 36°16′ |
| Altitude (Alt, m) | 982 | 0.3 | 1423 |
| Air temperature annual average (Temp, °C) | 11.2 | 8.3 | 17.7 |

**Table 1.** *Cont.*

| Variables | Median | Minimum | Maximum |
|---|---|---|---|
| Seasonality (Season, °C) | 19.2 | 15.8 | 22.4 |
| Precipitation total annual (Precip) | 611 | 355 | 1017 |
| Lake area (Area, ha) | 12.0 | 0.1 | 635 |
| Maximum depth (Depth, m) | 3.3 | 0.6 | 15.2 |
| Secchi depth (Sec, m) | 1.0 | 0.3 | 4.1 |
| Plant Volume Inhabited (PVI, %) | 6.9 | 0.0 | 79.9 |
| Chlorophyll a (Chl-a, µg/L) | 16.5 | 4.7 | 181.1 |
| Total phosphorous (TP, µg/L) | 72 | 18 | 402 |
| Total nitrogen (TN, µg/L) | 964 | 264 | 3250 |
| Salinity (Sal, ‰) | 0.30 | 0.06 | 14.50 |
| Species richness (number of fish species) | 3.5 | 1 | 11 |
| Shannon–Wiener diversity index (fish) | 0.62 | 0 | 1.9 |
| Fish (number of fish per net per night) | 53 | 1.5 | 1425 |
| Fish (gram fish per net per night) | 1119 | 116 | 4177 |
| Piscivorous fish proportion of total biomass | 0 | 0 | 0.9 |

*2.2. Fish Survey*

The dataset includes results from one fish survey in each lake conducted in summer (July and August) during the period 2006 to 2012 using Nordic benthic multi-mesh-size gillnets (length: 30 m; height: 1.5 m; 12 sections of 2.5 m with mesh sizes ranging from 5 mm to 55 mm knot to knot), the lowest mesh size of 5 mm allowing 0+ fish to be caught in the nets. The nets were placed in the littoral and pelagic zones parallel to the shore and left overnight. The number of nets per lake depended on lake area, ranging from two (one littoral and one pelagic) in lakes < 2 ha to eight in the largest lakes (>100 ha). Some studies have shown that multi-mesh-survey nets underestimate the number of small fish, e.g., [33], but Emmrich et al. [34] demonstrated strong correspondence between catches by multi-mesh gillnets and density calculated from hydroacoustic records.

In total, 50 fish species from 33 genera were caught in the 40 lakes. Species richness of the fish communities in each of the lakes ranged from 1 to 11 species, and the Shannon–Wiener diversity index (species diversity) ranged from close to 0 to 1.9 [31]. The number per unit effort (NPUE) calculated as catch per net per night ranged between 1.5 and 1425 ind./net. In most of the lakes, the proportion of piscivorous fish of the total fish biomass was low (Table 1). Further details about the fish community of these lakes can be found in Boll et al. [31].

The fork length of all fish caught was measured in the field. Fish without a forked caudal fin, for instance Cobitidae and Cyprinodontidae, were measured as total length. Fish length data were standardized by the geometric mean of the sample before the size diversity index, which is modified from the Shannon–Wiener diversity index for continuous data, and calculated (e.g., fish length data) [34].

Size diversity ($\mu$) was calculated as:

$$\mu = - \int\limits_{0}^{+\infty} p_x(x) \log_2 p_x(x)\,dx, \tag{1}$$

where $p_x(x)$ is the probability density function of the length of each individual fish. $p_x(x)$ was calculated using kernel estimation. This nonparametric approach is described in detail by [35] and has been successfully used on fish data by [36]. The size diversity index is the continuous analogue of the Shannon–Wiener diversity index and it is easy to interpret: a high size diversity means a wide size range and similar proportions of the different sizes along the size distribution [7,36]. The geometric mean length of fish and the number of size classes with 1 cm intervals were also used as a dependent variable.

### 2.3. Statistical Analyses

Generalized linear models (GLM) [37] were applied to assess the effect of the selected variables on fish size measures, i.e., geometric mean length, number of size classes and size diversity. Geometric mean length and size diversity were analysed using GLM with a Gaussian error distribution and an identity link function, while the number of size classes was analysed using GLM with a Poisson error distribution and a logarithmic link function.

As explanatory variables, the models included a subset of geo-climatic and other environmental variables selected. These variables were selected by use of a selection procedure including a Spearman correlation matrix (Table 2) and the variance inflation factor (VIF) to avoid including redundant (strongly correlated) variables in the models. A correlation factor > 0.6 was considered strong, and from each correlation pair only the variable with the lowest VIF was used in the analyses. Each subset included latitude, longitude, lake area, max depth, Secchi depth, salinity, PVI, and either of the correlated pairs: altitude or temperature, and precipitation or seasonality. As Chl-a was strongly correlated with both TP and TN, each subset included either Chl-a or both TP and TN. This procedure was also used by Boll et. al. [31]. Before the analysis, explanatory variables with a skewness ≥ 0.9 were log-transformed (i.e., lake area, max depth, salinity, TN, TP, and Chl-a: $\log_{10}(x)$; PVI: $\log_{10}(x + 1)$). The remaining variables had skewness values in the range from −0.54 to 0.73 and were not transformed.

Additionally, we investigated how the traditional taxonomic fish community measures were related to size diversity and number of size classes by including species richness and species diversity as explanatory variables in the models, as a second step in the analyses.

For each fish size variable, the full model was calculated, and the variation explained by each GLM was given as either adjusted $R^2$ or adjusted pseudo $R^2$. For geometric mean length and size diversity, which showed Gaussian error distributions, we calculated the adjusted $R^2$. For number of size classes, assuming Poisson error distribution, the variation explained by each GLM was estimated by calculating an adjusted pseudo $R^2$: $1 - ((\text{residual deviance} + k \times \varphi)/\text{null deviance})$, where k is the number of explanatory variables and $\varphi$ is the dispersion parameter. This pseudo $R^2$ calculation is adjusted for potential over- or under-dispersion in accordance with Heinzl and Mittlböck (2003) [38]. The dispersion parameter, $\varphi$, can be estimated by the generalized Pearson statistic, $\chi^2$, divided by the degrees of freedom, i.e., $\varphi = \chi^2/(n - k - 1)$ [38]. Furthermore, to evaluate the relative importance of each explanatory variable in explaining the variation in each fish size variable, modified Akaike information criterion for small sample sizes (AICc) was used. AICc was calculated for each submodel derived from the full model of each fish size variable. Subsequently, the relative importance of explanatory variables was estimated by summing the normalized model likelihoods ("Akaike weights") for each explanatory variable across all submodels in which the respective variable occurred. Thus, the larger the sum, the more important was the variable compared to other variables [39]. Calculations were done using the R package "MuMIn" [40]. All statistical analyses were carried out using R version 3.6.3 [41].

**Table 2.** Spearman correlation matrix including H′ (Shannon–Wiener species diversity), nSP (species richness), and relevant environmental and climatic variables (for remaining abbreviations, see Table 1). The correlation factors and *p* values are shown in the upper and lower part of the table, respectively. *n* = 39 lakes. Correlation factors for correlations ≥ 0.6 are shown in bold, indicating that these variables are strongly correlated; thus, only one of them should be included in the models.

|  | H′ | nSP | Lat | Lon | Alt | Area | Depth | Sec | Chl-a | TP | TN | Sal | PVI | Precip | Temp | Season |
|---|---|---|---|---|---|---|---|---|---|---|---|---|---|---|---|---|
| **H′** |  | **0.65** | 0.25 | −0.06 | **−0.49** | 0.26 | −0.30 | −0.06 | −0.10 | 0.09 | −0.08 | 0.24 | 0.05 | 0.22 | 0.30 | −0.54 |
| **nSP** | 0.000 |  | 0.31 | −0.07 | **−0.66** | **0.60** | −0.40 | −0.01 | 0.04 | 0.17 | 0.01 | 0.39 | 0.25 | −0.08 | 0.52 | −0.29 |
| **Lat** | 0.133 | 0.059 |  | 0.09 | −0.32 | −0.02 | −0.27 | 0.28 | −0.38 | −0.08 | −0.54 | −0.11 | 0.26 | 0.01 | −0.07 | −0.36 |
| **Lon** | 0.734 | 0.663 | 0.602 |  | 0.24 | −0.07 | −0.10 | 0.13 | 0.19 | 0.26 | −0.04 | 0.19 | 0.37 | −0.41 | −0.34 | 0.38 |

**Table 2.** *Cont.*

| | H' | nSP | Lat | Lon | Alt | Area | Depth | Sec | Chl-a | TP | TN | Sal | PVI | Precip | Temp | Season |
|---|---|---|---|---|---|---|---|---|---|---|---|---|---|---|---|---|
| **Alt** | 0.002 | **0.000** | 0.044 | 0.144 | | −0.32 | 0.42 | −0.09 | 0.15 | 0.07 | 0.06 | −0.49 | −0.07 | −0.03 | **−0.80** | 0.51 |
| **Area** | 0.116 | **0.000** | 0.924 | 0.690 | 0.045 | | −0.25 | −0.11 | 0.29 | 0.26 | 0.22 | 0.35 | 0.22 | 0.00 | 0.42 | 0.03 |
| **Depth** | 0.062 | 0.012 | 0.099 | 0.545 | 0.008 | 0.128 | | −0.57 | 0.03 | −0.26 | −0.07 | −0.35 | −0.55 | 0.22 | −0.43 | 0.07 |
| **Sec** | 0.722 | 0.961 | 0.081 | 0.415 | 0.573 | 0.487 | 0.000 | | −0.39 | −0.20 | −0.29 | 0.23 | 0.48 | 0.04 | 0.08 | −0.01 |
| **Chl-a** | 0.545 | 0.803 | 0.018 | 0.246 | 0.370 | 0.075 | 0.843 | 0.013 | | **0.71** | **0.74** | 0.21 | 0.09 | −0.29 | 0.14 | 0.33 |
| **TP** | 0.596 | 0.300 | 0.624 | 0.113 | 0.693 | 0.106 | 0.107 | 0.213 | **0.000** | | 0.56 | 0.16 | 0.32 | −0.23 | 0.04 | 0.23 |
| **TN** | 0.631 | 0.943 | 0.000 | 0.820 | 0.710 | 0.185 | 0.671 | 0.075 | **0.000** | 0.000 | | 0.21 | 0.07 | −0.13 | 0.26 | 0.29 |
| **Sal** | 0.138 | 0.015 | 0.500 | 0.239 | 0.001 | 0.030 | 0.028 | 0.161 | 0.198 | 0.330 | 0.196 | | 0.14 | −0.25 | 0.43 | 0.10 |
| **PVI** | 0.753 | 0.130 | 0.105 | 0.022 | 0.661 | 0.185 | 0.000 | 0.002 | 0.585 | 0.047 | 0.685 | 0.392 | | −0.18 | 0.21 | 0.10 |
| **Precip** | 0.187 | 0.644 | 0.935 | 0.010 | 0.849 | 0.993 | 0.174 | 0.808 | 0.078 | 0.162 | 0.419 | 0.118 | 0.282 | | 0.09 | **−0.69** |
| **Temp** | 0.067 | 0.001 | 0.676 | 0.035 | **0.000** | 0.008 | 0.007 | 0.649 | 0.407 | 0.813 | 0.114 | 0.007 | 0.194 | 0.579 | | −0.37 |
| **Season** | 0.000 | 0.072 | 0.025 | 0.016 | 0.001 | 0.879 | 0.672 | 0.963 | 0.039 | 0.167 | 0.076 | 0.563 | 0.548 | **0.000** | 0.019 | |

## 3. Results

Across lakes, the length of fish ranged from 3 cm for *Tilapia zillii*, *Aphanius anatoliae*, *A. danfordii* and *Gambusia holbrooki* to 48 cm for *Cyprinus carpio* and *Salmo trutta abanticus*. Geometric mean length of the fish ranged from 3.8 to 17.8 cm, with TN (negative impact) and altitude (positive impact) being the most important variables explaining variation (Table 3).

Size diversity in 39 of the 40 study lakes ranged from 0.92 to 2.73, the number of size classes being 4 to 25 per lake. The remaining lake had a clearly lower size diversity value of −0.460, and, though negative values may occur [35], it was considered an outlier (Bonferroni Outlier test, $p < 0.01$) and excluded from the subsequent analyses. This particular lake had a fish community dominated by *Alburnus escherichii* and a few *Tinca tinca* and *Cyprinus carpio*; however, 89% of all the fish belonged to the size class 9–10 cm, despite a total number of size classes of 13 covered in several of the other study lakes.

Size diversity and species diversity were strongly correlated (Pearson, $n = 39$: $r = 0.65$, $p < 0.0001$), and, when including species diversity in the GLM, the explained variation increased to 56%, with species diversity being the most important explanatory variable followed by precipitation, positively correlated (Table 3). The lakes with the highest size diversity and species diversity were characterized by species with little size overlap, occurring in relatively equal abundances, e.g., lakes with 11 species including small-bodied *Aphanius danfordii* and *Gambusia holbrooki*, medium-sized *Alburnus derjugini*, *Petroleuciscus borysthenicus* and *Pseudorasbora parva*, as well as larger-sized *Sander lucioperca* and *Cyprinus carpio* (Figure 2). By contrast, lakes with low size diversity were often strongly dominated by one species occurring in high numbers in one specific size class (e.g., *Rhodeus amarus* and *Aphanius anatoliae*), and/or exhibited low species richness (Figure 2). Size diversity also correlated positively with species richness (Spearman, $n = 39$: $r = 0.42$, $p = 0.0071$). The number of size classes also correlated positively with species diversity and species richness (Spearman, $n = 39$: $r = 0.41$, $p = 0.0089$ and $r = 0.57$, $p = 0.0002$, respectively), and in the GLM species diversity was retained together with lake area (positive impact) and salinity (negative impact) (Table 3). The most important variables, while for geometric mean length, were TN (negative impact) and altitude (positive impact) (Table 3).

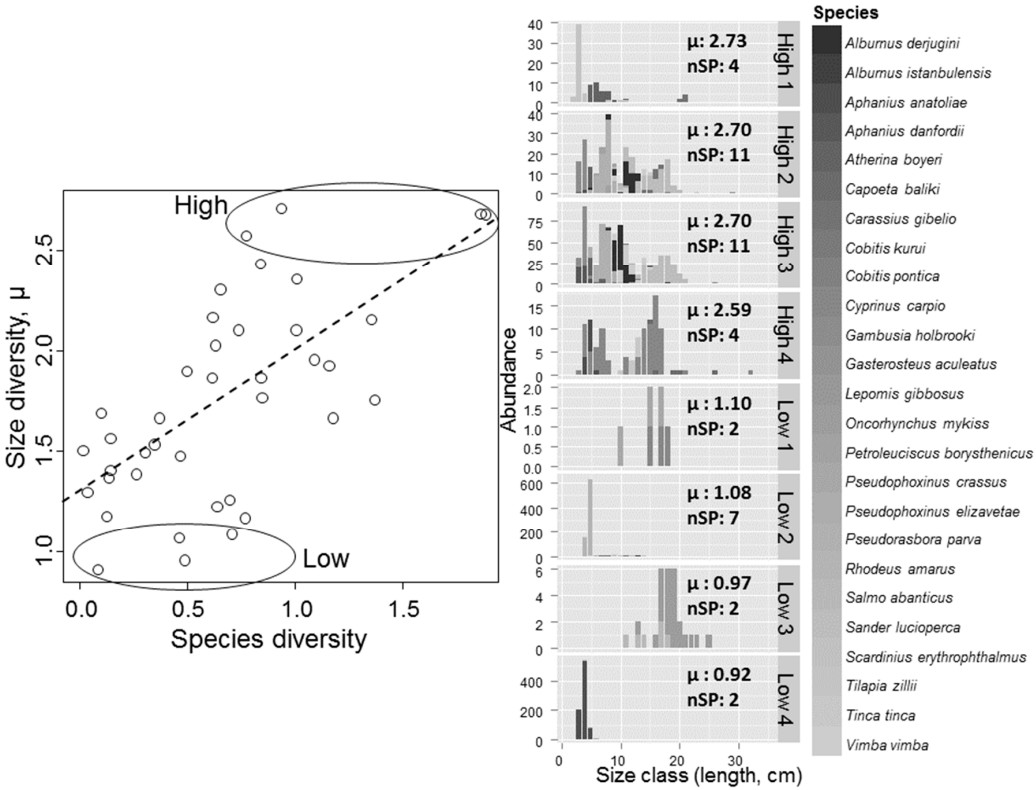

**Figure 2.** Correlation plot of size diversity (μ) and Shannon–Wiener species diversity for 39 lakes. Lakes with the highest and lowest size diversity are marked; to the right: histogram with characteristics of the fish community in each of the marked lakes. "High 1" is the lake with the highest size diversity and "Low 4" is the lake with the lowest size diversity. "nSP" is number of species. Regression (Pearson: $r = 0.65$, $p < 0.0001$) is given as a punctured line.

**Table 3.** Estimates from GLM analyses including data from 39 lakes. SD: size diversity; nSC: number of size classes; Length.geom: geometric mean length. For remaining abbreviations, see Tables 1 and 2. Lake area, max depth, salinity, TN, TP, and Chl-a were log10(x)-transformed. H' and PVI were log10(x + 1)-transformed. Geometric mean length did not correlate with species diversity, and, thus, a second step was not included for this variable. The sum of weights is shown as the relative importance of each variable (Rel. import).

| | (Intercept) | H' | Lat | Lon | Alt | Precip | Area | Depth | Sec | PVI | TP | TN | Sal |
|---|---|---|---|---|---|---|---|---|---|---|---|---|---|
| **SD excl. H'** | | −0.7775 | −0.0293 | 0.0764 | −0.0003 | 0.0017 | 0.1060 | −0.1354 | 0.2351 | −0.1819 | −0.1895 | 0.3911 | −0.0740 |
| **Rel. import.** | | | 0.28 | 0.56 | 0.62 | 0.98 | 0.28 | 0.24 | 0.20 | 0.24 | 0.21 | 0.30 | 0.27 |
| *AICc: 70.0;* | *Adj. R²: 0.22;* | | | | | | | | | | | | |
| **SD incl. H'** | −1.6612 | 2.7494 | −0.0251 | 0.0514 | −0.0001 | 0.0009 | 0.1378 | 0.0973 | 0.7644 | −0.2292 | −0.3127 | 0.6002 | −0.2005 |
| **Rel. import.** | | 1.00 | 0.28 | 0.42 | 0.18 | 0.91 | 0.27 | 0.19 | 0.34 | 0.30 | 0.47 | 0.54 | 0.19 |
| *AICc: 54.6;* | *Adj. R²: 0.56;* | | | | | | | | | | | | |

| | (Intercept) | H' | Lat | Lon | Alt | Precip | Area | Depth | Sec | PVI | TP | TN | Sal |
|---|---|---|---|---|---|---|---|---|---|---|---|---|---|
| **nSC excl. H'** | 1.331 | | 0.0356 | 0.0036 | −0.0006 | 0.0000 | 0.1966 | −0.0916 | −0.1731 | 0.0135 | −0.0855 | −0.0265 | −0.1315 |
| **Rel. import.** | | | 0.32 | 0.21 | 0.24 | 0.21 | 0.99 | 0.21 | 0.23 | 0.22 | 0.21 | 0.26 | 0.62 |
| *AICc: 270.9;* | *Adj. pseudo R²: −0.16;* | | | | | | | | | | | | |
| **nSC incl. H'** | 1.034 | 2.2934 | 0.0378 | −0.0202 | 0.0002 | −0.0007 | 0.2407 | 0.0714 | −0.2222 | −0.0249 | −0.2168 | 0.1043 | −0.2277 |
| **Rel. import.** | | 1.00 | 0.24 | 0.22 | 0.54 | 0.41 | 0.98 | 0.21 | 0.22 | 0.19 | 0.37 | 0.21 | 0.79 |
| *AICc: 252.5;* | *Adj. pseudo R²: 0.28;* | | | | | | | | | | | | |

| | (Intercept) | | Lat | Lon | Alt | Precip | Area | Depth | Sec | PVI | TP | TN | Sal |
|---|---|---|---|---|---|---|---|---|---|---|---|---|---|
| **Geom. length** | 17.7879 | | 0.4004 | −0.0957 | 0.0031 | 0.0001 | −1.0663 | −2.3956 | −6.6931 | 0.2815 | −0.5769 | −4.4252 | −0.2741 |
| **Rel. import.** | | | 0.30 | 0.20 | 0.70 | 0.20 | 0.49 | 0.24 | 0.56 | 0.22 | 0.21 | 0.80 | 0.46 |
| *AICc: 225.7;* | *Adj. R²: 0.28* | | | | | | | | | | | | |

## 4. Discussion

Size diversity of lake fish communities in the Turkish lakes studied related strongly to species diversity, and the number of size classes and, although less strongly, to species richness, suggesting that these size metrics provide valuable integrated information on the fish community structure and functionality of the lakes. These findings support other studies revealing that size [6,42,43] and morphological [44,45] measures can be complementary to or substitute traditional taxonomic diversity measures. The lakes with the highest size diversity of fish communities were also those with highest species diversity and hosted several species with little size overlap, which suggests that these species have different niches and functional roles, probably through strong completion that resulted in niche-packing, as also observed in some highly diverse marine fish communities [22,24]. These results contrast with findings from northern European lakes, where fish communities tend to comprise only one or a few species, but each of which exhibits a wide range in size (i.e., communities with high size diversity but low species diversity) [11].

The strong positive correlation between size diversity and species diversity shows that fish communities with high species diversity (i.e., high species richness and evenness) also support a large range of size classes with more even distribution among size classes. These lakes included both small-bodied species like *Aphanius danfordii* and *Gambusia holbrooki*, as well as larger-sized species like *Sander lucioperca* and *Cyprinus carpio*. On the contrary, lakes with few species, and especially lakes strongly dominated by one size class (or cohort), like the lakes dominated by *Rhodeus amarus* or *Aphanius anatoliae*, had low size diversity (Figure 2). This pattern was even more pronounced in the outlier lake where a single size class strongly dominated the fish community with a resultant low size diversity. The peculiarity of the fish community in this lake was not detected by traditional taxonomic diversity measures (i.e., species richness), nor reflected in number of size classes (nSC = 13), even though it likely had a strong influence on community functioning, emphasizing that size measures may provide additional information about functioning to the traditional methods.

Although species diversity was the most important explanatory variable in the GLMs for size diversity and number of size classes, a few climatic and environmental variables also remained important when including species diversity to the GLM, emphasizing the potential use of size metrics as ecological indicators. For size diversity, precipitation was included, while for number of size classes, it was lake area and salinity. Generally, lakes with low fish size diversity were found in areas with low precipitation and large seasonal differences in temperature. This is in line with Schleuter et al. [46], who showed that high temperatures and low precipitation in areas subjected to geographic isolation result in low functional richness, and extinction rates are likely higher than rates of re-colonization or speciation rates in such environments. The findings also agree with a study at the European scale showing smaller fish body sizes in lakes exhibiting greater variations in temperature and low precipitation [12], maybe because small body size can be an advantage for fish inhabiting strongly seasonal environments [47]. The physiological tolerance hypothesis [48] also predicts lower taxonomic richness in climatically less suitable areas. In our study, lakes experiencing low precipitation and large seasonal differences in temperature were often, but not exclusively, situated in the central Anatolian Plateau which, in addition to providing less favorable climatic conditions, exhibit dispersal limitations and supports less species than coastal lowland lakes [31]. However, our results suggest that precipitation and/or seasonality affect size diversity more than just through changes in species diversity, which may be explained by loss of fish cohorts (e.g., young of the year or old fish) rather than loss of fish species in years with especially hard conditions (e.g., drought, or extreme high or low water temperatures). In such cases, size diversity will provide complementary information to species diversity about lake stability.

Lake area remained important and had a positive impact on number of size classes even when species diversity was included in the models. A larger area of lakes may have a direct positive impact on the number of size classes, as the larger the area of lakes, the

more diverse habitats probably are, potentially promoting the co-existence of fish in more size classes [21], and may be inhabited by larger fish [14,49]. Emmrich et al. [36] have previously found a positive impact of lake area on size diversity based on a set of European lakes > 50 ha, but compared to other variables, this impact on size diversity appeared less important in our relatively small lakes (25 out of 39 lakes were ≤ 50 ha). Total nitrogen and altitude were found to be the most important variables explaining geometric mean length of the fish in the lakes: it decreased with increasing TN and increased with increasing altitude. These findings comply with former studies showing warmer and more eutrophic lowland lakes being dominated by small-sized fish [12,13,17,30].

Salinity contributed (negatively) to the variation in size diversity, number of size classes and, slightly, to geometric mean length. In a study of 24 studied lakes in Xinjiang province, China, Vidal et al. [50] also found that fish size diversity was negatively correlated to salinity and positively to lake area, and the changes in size diversity seemed to be mediated by salinity effects on species diversity, like in our study. Fish communities in eutrophic brackish lakes are often also dominated by small-bodied fish species [7,13,51,52].

As size diversity is related to species diversity and species richness, the size measures might also be impacted by the introduction of new fish species. Many Turkish lakes support alien or translocated species [31,53]. The introduced species included both small-sized (e.g., *Gambusia holbrooki*) and large-sized (e.g., *Sander lucioperca*) species, and they were found both in lakes with low and high size diversity. However, a preliminary data analysis (not included) showed that introduced species did not correlate with any of the sizes measures and, accordingly, they were therefore not included as explanatory variables.

## 5. Conclusions

Importantly, we found that fish size diversity and species diversity were strongly correlated, and fish size diversity in lakes in western Türkiye was strongly related to total annual precipitation, while the number of size classes increased with increasing lake area and decreased with increasing salinity. The geometric mean length of fish decreased with increasing TN concentration. Therefore, these size metrics may be useful ecological and biodiversity indicators, integrating variations at both intraspecific and interspecific levels, that can complement the traditional taxonomic diversity measures (as found in other studies, e.g., Brucet et al. [10,11]), but not least in areas where climate change may lead to reduced precipitation and increased salinization, such as in Türkiye [54] and other Mediterranean climatic regions.

**Author Contributions:** The sampling of lakes was designed by M.B. and E.J. and executed by M.B., Ş.E., N.F., A.Ö., Ü.A.B. and E.E.L. The study was conceptualized by T.B., S.B., E.J. and M.B.; T.B. did the analyses and wrote up the first draft. M.B., E.J., S.B. and N.F. revised the paper. All authors have read and agreed to the published version of the manuscript.

**Funding:** This study was supported by Scientific and Technological Research Council of Türkiye (TÜBİTAK) projects (Creating Adaptation and Mitigation Strategies by Determining the Structural Role and Development of Aquatic Plants in the Shallow Lakes in the Mediterranean Climate Belt and Determining the Factors Affecting Their Development in the Past, Today and Warmer Conditions, project no: ÇAYDAG-105Y332 and Determining the Factors Affecting the Structural Role and Development of Aquatic Plants in the Shallow Lakes in the Mediterranean Climate Zone and Creating Adaptation and Mitigation Strategies by Determining in the Past, Today and Warmer Conditions, project no: ÇAYDAG-110Y125), the Middle East Technical University (METU)- Scientific Research Projects (BAP) program of Türkiye (project no: BAP.07.02.2009-2012), FP-7 REFRESH Project (Adaptive strategies to Mitigate the Impacts of Climate Change on European Freshwater Ecosystems, Contract No.: 244121) and EU-7th Framework MARS project (Managing Aquatic ecosystems and water Resources under multiple Stress), (Theme 6, Environment including Climate Change, Contract No.: 603378. T.B. was supported by the TÜBİTAK, program 2216—Research Fellowship Program for Foreign Citizens (Ref: B.14.2.TBT.0.06.01.03-216.01-24962). S.B.'s contribution was supported by the TÜBİTAK, program 2221—Visiting Scientist Fellowship Program, by the Marie Curie Intra European Fellowship (CLIMBING, No. 330249) and by the Spanish Ministry of Science, Innova-

tion and Universities (grant number: RTI2018-095363-B-I00). E.J. and M.B. were also supported by the TÜBITAK program 2232 (Climate change effects on trophic structure and dynamics in saline and brackish water based on a space-for-time field sampling, controlled mesocosm experiments, paleoecology, remote sensing and modelling, CLIM-SALTLAKES, project no: 118C250) and the H2020 EU-funded project AQUACOSM-plus (Network of Leading European AQUAtic MesoCOSM Facilities Connecting Mountains to Oceans from the Arctic to the Mediterranean, Contract No.: 365 871081). Furthermore, M.B. and S.B. were supported by H2020 EU-funded project PONDERFUL project (Pond Ecosystems for Resilient Future Landscapes in a Changing Climate, Grant No. 869296).

**Data Availability Statement:** No new data were created or analyzed in this study. Data sharing is not applicable to this article.

**Acknowledgments:** We thank Müfit Özuluğ, Ayşe İdil Çakıroğlu, Ülkü Nihan Tavşanoğlu, Gülşah Saç, Çiğdem Kaptan, Gürçay Kıvanç Akyıldız for assistance in the field and Anne Mette Poulsen for valuable editing of the manuscript.

**Conflicts of Interest:** The authors declare no conflict of interest.

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
