# Peer review of "Fish Size Structure as an Indicator of Fish Diversity: A Study of 40 Lakes in Türkiye"

_water, doi:10.3390/w15122147_

Round 1
Reviewer 1 Report
Comments are attached in the file
Specific comments:
1. Do you think only one sampling is enough for diversity study?
2. Are you sure you have taken all the length group of the available fishes, it is some times very difficult to get all the length groups of a single species in a single sampling season.

Author Response
Reviewer 1.
Specific comments:
- Do you think only one sampling is enough for diversity study?
Reply: Clearly, it would be better to have multiple samples and even preferably have monitoring which expands over several years but then it is not possible to monitor large number of lakes. Thus, space-for-time-sampling approach with employing the same protocol during the peak of the growing seasons can provide data from large number of lakes which vary in latitude, physio-chemical conditions that allow relating the fish richness to the given conditions. This is what this study aimed at and have been commonly used in other studies of various biota in the past (Pickett, 1989; Olesen et al., 2010; Meerhoff et al., 2012; BeklioÄŸlu et al., 2020; Monteiro et al., 2022).
BeklioÄŸlu, M., Bucak, T., Levi, E. E., ErdoÄŸan, Åž., Özen, A., Filiz, N., Bezirci, G., ÇakıroÄŸlu, A. Ä°., TavÅŸanoÄŸlu, N., Gökçe, D., Demir, N., ÖzuluÄŸ, M., Duran, M., Özkan, K., Brucet, S., & Jeppesen, E. (2020). Influences of climate and nutrient enrichment on the multiple trophic levels of Turkish shallow lakes. Inland Waters.
https://doi.org/10.1080/20442041.2020.1746599
Meerhoff, M., Teixeira-de Mello, F., Kruk, C., Alonso, C., Gonz??lez-Bergonzoni, I., Pacheco, J. P., Lacerot, G., Arim, M., Beklio??lu, M., Brucet, S., Goyenola, G., Iglesias, C., Mazzeo, N., Kosten, S., & Jeppesen, E. (2012). Environmental Warming in Shallow Lakes. A Review of Potential Changes in Community Structure as Evidenced from Space-for-Time Substitution Approaches. In Advances in Ecological Research (Vol. 46). https://doi.org/10.1016/B978-0-12-396992-7.00004-6
Monteiro, M. R., Marshall, A. J., Hawes, I., Lee, C. K., McDonald, I. R., & Cary, S. C. (2022). Geochemically Defined Space-for-Time Transects Successfully Capture Microbial Dynamics Along Lacustrine Chronosequences in a Polar Desert. Frontiers in Microbiology. https://doi.org/10.3389/fmicb.2021.783767
Olesen, J. M., Dupont, Y. L., O’Gorman, E., Ings, T. C., Layer, K., Melián, C. J., Trøjelsgaard, K., Pichler, D. E., Rasmussen, C., & Woodward, G. (2010). From Broadstone to Zackenberg. Space, Time and Hierarchies in Ecological Networks. In Advances in Ecological Research. https://doi.org/10.1016/B978-0-12-381363-3.00001-0
Pickett, S. T. A. (1989). Space-for-Time Substitution as an Alternative to Long-Term Studies. In Long-Term Studies in Ecology. https://doi.org/10.1007/978-1-4615-7358-6_5
- Are you sure you have taken all the length group of the available fishes, it is sometimes very difficult to get all the length groups of a single species in a single sampling season.
Reply: We selected the period of the year where YOY fish could be caught and the set of mesh size used should give a good description of the size distribution (CEN, 2005).
Other Comments
- Why size distribution is not a factor in the correlation estimation?
Because the size distribution is our response variable and here we only show correlations for the predictors. The variation of the response variable in relation with predictors is tested by using GLMs.
- What if a single size classes but different species (e.g. small indigenous fishes in the wetlands), practically the diversity is high but in the present method the diversity will be less?
Reply: The reviewer refers to a situation with different fish species with the same size (thus high species diversity but low size diversity). This situation is indeed possible but it was not observed in this study (overall we found a good correlation between size diversity and species diversity).
- The single size classes is impposible, because different mesh size and different age group was targeted in the survey, then how single size classes only were recorded??
Reply: We agree it is highly unlikely to find a lake with only one size class of fish. This was not the case in our study either but we had a lake were one size class strongly dominated the fish community. We found in total 13 size classes in the lake but one size class (from a single species) was found in very high numbers compared to the other size classes. In total three fish species were found in the lake.
- It (including species diversity to GLM) can be used as pseudo indicator with clearly mentioned assumptions, when it will not work.
Reply: In our GLM model some climatic and environmental variables are significantly related to size diversity, even when controlling for the effect of species diversity, this suggests that size diversity can be an indicator of climatic and environmental variation.
- But brackish water diversity is always higher than freshwater and marine
Reply: Yes, this is the case but in this study we compared the freshwater lakes which already have high diversity including endemic species with inland saline lakes in which the salinization creates an extra stressors.
Reviewer 2 Report
I made some corrections, remarks, and suggestions on the attached MS. Taking them into account might result in a higher scientific value.

Author Response
Reviwer 2:
I made some corrections, remarks, and suggestions on the attached MS. Taking them into account might result in a higher scientific value.
Reply: Thank you for your comments, we took them into account and make the changes seen below.
- Some more details about the sampled lakes would be informative (in a new table?). Perhaps also a map showing their location (as in the cited Boll et al., 2016 or Beklioglu et al., 2020).
Reply: Now we added more details to Table 1 and also a map which shows the sampled lakes, names, number and how many of them are hydrologically connected.
- It is more than funny that the number of fish nor the sampling is not even mentioned.
Reply: We have now added information on species richness and diversity as well as NPUE (number of fish caught per unit effort) and proportion of piscivorous fish biomass. We have made minor adjustments to the description of sampling method.
- This part could be the Conclusions.
Reply: “conclusion” heading has been added.
Reply: Other minor typos and references were also corrected.
Reviewer 3 Report
Review of Fish size structure as an indicator of fish diversity: a study of 40 lakes in Türkiye
This paper evaluates 40 lakes to understand the relationship between size structure and fish diversity. This is very important to help with monitoring changing systems especially climate change and urban growth. The paper captures fish data using mesh nets to sample the fish populations. The analysis is based on a GLM and includes some environmental data.
In general I liked the paper and it was well written. The references need fixing up with some mistakes (33/34 & 49 for example) but overall the text was clear. My comments are minor:
1. Table 1 shows the factors being used. This list is limited to many extents. Lakes are self contained in many respects and this needs to be recognised. Factors such as the lake’s geomorphological history, geospatial linkages to rivers, Lake catchment size, depth profile (not just max depth), lake catchment urban %, fishing pressure and temporal changes in recent history are some that are important.
2. Formulae 1 has the line numbers within the formulae but this will be sorted by an editor.
3. Line 152 GLM is a blunt instrument for this sort of analysis. Describe how the authors are confident with a linear approach. Was a more flexible approach like Random Forests considered? Was an analysis of the individual weights conducted to determine if some key lakes were critical in the significance estimation (more than line 201)?
4. In Table 1 include the acronyms and then take out from table 2 and 3 in table descriptions.
5. A map of the lakes/ sample sites is needed with relevant geomorphological structures. Are any of these lakes artificial? Are any of the lakes linked together by rivers?
6. Line 360 Introduced fish can be a major problem for similar studies in Africa. Can the disruption to the trophic structure be discussed. Is there a common fish biota that occurs in all lakes? Are there other introductions such as invasive water weeds? Or fish like eels or molluscs that were not sampled?
7. Fishing pressure is a glaring omission here and a critical aspect of the paper. Some sort of measure is required. Similarly for Total Nitrogen and Altitude, these factors seem to be surrogates for other more direct factors such as % farming and water temperature profiles. Is dissolved oxygen a factor?
8. Is the July & August sampling the most appropriate time? Are there seasonal constraints here?
9. Line 123 the sampling period covers 6 years but there is little exploration of the temporal variation in the data.
1 Line 189 The AIC is a good approach but the methods and results need to highlight: The number of parameters in the model, the information score of the model, the difference in AIC score between the best models, the model weights, the sum of the AICc weights and the Log-likelihood. This was not reported in the results.
1 Is there a spatial correlation influence?
Author Response
This paper evaluates 40 lakes to understand the relationship between size structure and fish diversity. This is very important to help with monitoring changing systems especially climate change and urban growth. The paper captures fish data using mesh nets to sample the fish populations. The analysis is based on a GLM and includes some environmental data.
In general, I liked the paper and it was well written. The references need fixing up with some mistakes (33/34 & 49 for example) but overall the text was clear. My comments are minor:
Reply: We are grateful the comments of the reviewer and we will address them one by one below. We have corrected the mistakes concerning references.
- Table 1 shows the factors being used. This list is limited to many extents. Lakes are self contained in many respects and this needs to be recognised. Factors such as the lake’s geomorphological history, geospatial linkages to rivers, Lake catchment size, depth profile (not just max depth), lake catchment urban %, fishing pressure and temporal changes in recent history are some that are important.
Reply: We totally agree with reviewer on the suggestions. Thus we expanded the Table 1 as much as possible. Extra variables were added as requested. We also included a new figure which shows the lakes on the Türkiye map. Furthermore, we included info on which of the sampled lakes are hydrologically connected. Unfortunately we have no information on lake catchment size, depth profile, lake catchment urban %, fishing pressure.
- Formulae 1 has the line numbers within the formulae but this will be sorted by an editor.
Reply: Thanks for noticing, we will handle this with the editor.
- Line 152 GLM is a blunt instrument for this sort of analysis. Describe how the authors are confident with a linear approach. Was a more flexible approach like Random Forests considered? Was an analysis of the individual weights conducted to determine if some key lakes were critical in the significance estimation (more than line 201)?
Reply: There are some reasons why we choose GLMs over Random Forests. First of all, GLMs offer greater interpretability compared to Random Forests by providing explicit parameter estimates and statistical inference, allowing us to understand the magnitude and direction of the explanatory variables on response variables which is important for studying environmental or ecological phenomena in lakes. Secondly, GLMs tend to be simpler models than Random Forests, which can be advantageous when working with a small dataset. With only 40 lakes, the number of observations might be limited, making it easier to overfit complex models like Random Forests. Finally, with GLMs we can test the interactions which we can not do this with RF analyses.
When it comes to linearity, we used one of the advantages of GLMs is that they can handle a wide range of response distributions, beyond just the linear Gaussian distribution. GLMs allow us to specify a link function that connects the linear predictor (the combination of predictors and their coefficients) to the response variable. This link function helps to transform the response variable so that it is compatible with the assumed distribution of the response in the GLM. We changed the link function according to our models’ distributions. On the other hand, Random Forests, being non-parametric models, do not rely on explicit distributional assumptions but might not capture the specific characteristics of the response variable as effectively.
Additionally, Figure 1 shows the lakes with highest and lowest size diversity and in this case the lakes that potentially could be most critical in the significance estimation. However, in this individual approach we do not find any of the lakes critical to an extend where the results are not trustworthy.
- In Table 1 include the acronyms and then take out from table 2 and 3 in table descriptions.
Reply: Thank you for your suggestion. We added acronyms in Table 1, and the additional ones were added to Table 2 and 3.
- A map of the lakes/ sample sites is needed with relevant geomorphological structures. Are any of these lakes artificial? Are any of the lakes linked together by rivers?
Reply: We have added a map of Türkiye on which the sampled lakes were added on. We also have shared an information on which of the study lakes are hydrologically connected. Furthermore, all the lakes sampled are natural lakes that is added to the text.
- Line 360 Introduced fish can be a major problem for similar studies in Africa. Can the disruption to the trophic structure be discussed. Is there a common fish biota that occurs in all lakes? Are there other introductions such as invasive water weeds? Or fish like eels or molluscs that were not sampled?
Reply: Introduced species are included in the dataset and discussed in details in Boll et al., 2016. Therefore, we are not discussing this topic in this paper.
- Fishing pressure is a glaring omission here and a critical aspect of the paper. Some sort of measure is required. Similarly for Total Nitrogen and Altitude, these factors seem to be surrogates for other more direct factors such as % farming and water temperature profiles. Is dissolved oxygen a factor?
Reply: In Türkiye, the lakes are owned by the state and fishing with a net is forbidden unless it is necessary and then it requires a license. Whereas angling is very selective toward carp. We have no info on fishing pressure on the lakes, but given the above, we expect it is of minor importance.
- Is the July & August sampling the most appropriate time? Are there seasonal constraints here?
Reply: These two months are the peak growing season and we believe that they are the appropriate time to reflect the biomass production. Moreover, it is the time when YOY fish are caught in the nets.
- Line 123 the sampling period covers 6 years but there is little exploration of the temporal variation in the data.
Reply: We checked the mean annual air temperature within the years and did not observe a strong temporal variability between the years (see figure below), that otherwise could have influenced our results. Moreover, it is typical for these kind of studies to cover many years (as sampling is man-power demanding), the rationale being that year-to-year variations in individual lakes are small compared to the huge gradient covered by the many lakes and therefore of minor importance for the outcome of the analyses.
- Line 189 The AIC is a good approach but the methods and results need to highlight: The number of parameters in the model, the information score of the model, the difference in AIC score between the best models, the model weights, the sum of the AICc weights and the Log-likelihood. This was not reported in the results.
We have added the sum of the AICc weights to Table 3 as it was explained in the text (Lines 202-209). Since we used the full model that made it possible to compare it with the model given at Boll et al., 2016 (Ref no, 29), thus we do not need to add the parameters related to the model selection such as information score of the models, the difference in AIC score between the best models, and the Log-likelihood. We provide the estimates from the GLM models in Table 3 as the model weights and reordered the names of the variables. When it comes to number of the parameters in the models, since we used the full model for each response variable, the number of the parameters are the same and can be seen in Table 3.
- Is there a spatial correlation influence?
Reply: We did not test the spatial correlation influence for several reasons 1) the lakes wer generally well-distibuted in the entire western part of Türkiye (see Fig 1) and 2) there are many variables that potentially could be of higher importance (e.g., slope, distance to sea, climatic and topographic variabilities, land use, connectivity) besides the included latitude, longitude and area. Taken into account the relatively few lakes included and the large geographical scale covered, we find that it is not needed (overkill) to include spatial factors to elucidate the research questions raised in this paper.
Round 2
Reviewer 1 Report
I am fine with the revision